



# Comparison of BARRA and ERA5 in Replicating Mean and Extreme Precipitation over Australia

Kevin K. W. Cheung[1] kevin.cheung@nuist.edu.cn

Fei Ji[2,3] fei.ji@environment.nsw.gov.au

Nidhi Nishant[4] nidhi.nishant@transport.nsw.gov.au

Jin Teng[5] Jin.Teng@csiro.au

James Bennett[6] james.bennett@csiro.au

De Li Liu[7] de.li.liu@dpi.nsw.gov.au

[1]School of Emergency Management, Nanjing University of Information Science and Technology, Jiangsu 210044, China

[2] Science, Economics and Insights Division, NSW Department of Climate Change, Energy, the Environment and Water, Sydney, NSW 2150, Australia

[3] Australian Research Council Centre of Excellence for Climate Extremes, University of New South Wales, Sydney, Australia

[4]Climate Change Research Centre, University of New South Wales, Sydney, NSW, Australia

[5] CSIRO Environment, GPO Box 1700, Canberra, ACT 2601, Australia

[6] CSIRO Environment, Research Way, Clayton, VIC 3168, Australia

[7]NSW Department of Primary Industries, Wagga Wagga Agricultural Institute, Wagga Wagga, NSW 2650, Australia

Corresponding author: Dr. Kevin Cheung, School of Emergency Management, Nanjing University of Information Science and Technology, Jiangsu 210044, China. Email: kevin.cheung@nuist.edu.cn



## Abstract

Reanalysis datasets are critical in climate research and weather analysis, offering consistent
historical weather and climate data crucial for understanding atmospheric phenomena, and
validating climate models. However, biases exist in reanalysis datasets that would affect their
applications under circumstances. This study evaluates BARRA, which is a high-resolution
reanalysis for the Australian region, and ERA5 in simulating mean precipitation and six
selected precipitation extremes for their climatology, temporal correlation, coefficient of
variation and trend. Both models reproduce spatial patterns of mean precipitation well with
minor biases. ERA5 shows stronger temporal correlations, superior inter-annual precipitation
accuracy, and lower biases in coefficient of variation compared to BARRA, especially in
Northern Australia. However, both models exhibit substantial biases in trend, underestimating
increasing trends in Northern Australia. ERA5 underestimates dry days and heavy rainfall,
while BARRA tends to overestimate these extremes. Temporal correlations for extreme
precipitation indices are weaker compared to mean annual precipitation. Notable differences
exist in variability biases, with BARRA showing larger biases, especially for heavy
precipitation in inland regions and Northern Australia. While both datasets replicate the main
trends, biases persist. Overall, the evaluation results support application of both datasets for
climatology analyses, but caution is advised for variability and trend analyses, particularly for
specific extremes.

**Key words:** BARRA, ERA5, extreme indices, temporal correlation, coefficient of variation,
trend

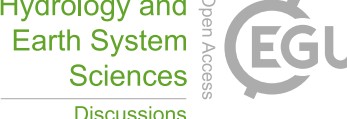

## 1. Introduction


Reanalysis dataset is created by combining historical observational data from various
sources, such as weather stations, satellites, buoys, and more, with modern data assimilation
techniques and numerical models (Kalnay, et al. 1996; Saha, et al. 2010; Dee et al. 2011;
Kobayashi et al. 2015, Poli et al. 2016; Hersbach 2020). The fundamental aim of reanalysis is
to construct a uniform and coherent historical archive of various atmospheric and
environmental parameters, such as temperature, humidity and wind patterns, on either a
regional or a global scale.
These datasets are invaluable for climate studies, weather analysis and model validation
as they provide a uniform representation of historical climate conditions. For instance,
Quagraine et al. (2020) used five global reanalysis datasets to investigate the variability of West
African summer monsoon precipitation, showing all datasets could represent the average
rainfall patterns and seasonal cycle.  Dai et al. (2023) utilized the fifth-generation European
Centre for Medium-Range Weather Forecasts (ECMWF) reanalysis (ERA5) data to estimate
rainfall erosivity on the Chinese Loess Plateau, finding rainfall erosivity derived from ERA5
was highly consistent with those derived from the meteorological stations. Cheung et al. (2023)
employed ERA5 to evaluate storm conditions in regional climate simulations, demonstrating
regional climate models can capture climatology of measurements of storm severity over land
including their spatial patterns and seasonality. Numerous studies have used reanalysis datasets
as inputs for regional climate models (RCMs) to evaluate the models' capability in replicating
observed climatic patterns (Solman et al., 2013; Ji et al., 2016; Fita et al., 2016, Di Virgilio et
al., 2019; Capecchi et al., 2023; Di Virgilio et al., 2024; Ji et al., 2024).
While reanalysis datasets provide valuable insights into historical weather and climate
conditions, they have limitations and uncertainties, given that they are modelled outputs rather
than direct observations. Many studies have evaluated reanalysis data across various variables





and regions. For instance, Betts et al. (2019) assessed ERA5 biases in near-surface variables
over Canada, highlighting its improved performance over ERA-Interim, though precipitation
biases remained significant. Similarly, Hu and Yuan (2021) and Jiang et al. (2021) found that
ERA5 precipitation accurately captured rainfall pattern over the Eastern Tibetan Plateau and
mainland China, but under-estimated intensity. Izadi et al. (2021) found ERA5 performed
better at monthly and seasonal timescales in Iran, underestimating coastal summer precipitation
and overestimating it in mountains. Jiao et al. (2021) and Qin et al. (2021) found ERA5
overestimated summer precipitation and frequency in China but underestimated intensity
during the warm season. Lei et al. (2022) and Shen et al. (2022) noted ERA5's limitations in
simulating extreme precipitation events in China, especially for high-end extremes.
Comparisons between reanalysis datasets have also been conducted. Wang et al. (2019)
found that both ERA5 and ERA-Interim exhibited warm biases over Arctic Sea ice, with larger
biases in cold season than warm season. Lei et al. (2020) showed ERA5 improved cloud cover
simulation over eastern China but not over the Tibetan Plateau, when compared to ERA-
Interim. Gleixner et al. (2020) found ERA5 reduced biases in temperature and precipitation
over East Africa compared to ERA-Interim but still struggled with long-term trends. Song and
Wei (2021) found both ERA5 and MERRA-2 captured night precipitation peaks over North
China, but only ERA5 accurately reflected the afternoon peak. Li et al. (2022) concluded that
ERA5 performed better than ERA-Interim, JRA55, and MERRA-2 in capturing precipitation
over the Poyang Lake Basin. A summary of the above literature review can be found in Table
S1.
In Australia, reanalyses like NCEP (Kalnay et al., 1996), JRA-55 (Kobayashi et al.,
2015), ERA-Interim (Dee et al., 2011), and ERA5 (Hersbach et al., 2020) are commonly used,
alongside the Australian Bureau of Meteorology's high-resolution (12 km) BARRA reanalysis.





BARRA covers Australia, New Zealand, and Southeast Asia (Su et al., 2019), while BARRA-
C offers even higher-resolution (1.5 km) analysis for four capital cities (Su et al., 2021).

May et al. (2021) found BARRA reliable, though it showed seasonal and diurnal biases.

Other studies, like Pirooz et al. (2021), compared BARRA with global reanalyses, concluding
BARRA performed better for precipitation and temperature in New Zealand but lagged behind
ERA5 for high gust winds. Du et al. (2023) used BARRA for estimating daily precipitation in
ungauged Australian catchments, while Hobeichi et al. (2023) employed BARRA to train
statistical models for downscaling. Acharya et al. (2019, 2020) found BARRA's precipitation
performance varied by region, with poorer results in tropical areas. Nishant et al. (2022)
suggested higher resolution in BARRA-C didn't always improve precipitation simulations,
while Choudhury et al. (2023) noted ERA5 performed better for mean temperatures than
extremes in Australia. These previous studies on BARRA and BARRA-C have also been
summarized in Table S1.

However, there is a gap in the existing studies concerning the intercomparison of

various reanalyses, such as BARRA and ERA5, specifically in relation to precipitation
extremes over Australia. In this study, we aim to bridge this gap by evaluating and comparing
the performance of BARRA and ERA5 in capturing precipitation extremes. While the
traditional evaluation methods focusing on climatology (long-term mean), here we also include
temporal correlation, coefficient of variation and trend in evaluation to quantify their overall
performance, which have not been examined before in previous studies. By assessing climate
means and extremes and quantifying their biases, this study provides a valuable reference for
selecting appropriate datasets for specific applications and cautions against treating reanalysis
data as observations. The paper is organized as follows: Section 2 introduces the reanalysis
datasets and observational data used for evaluation. Section 3 outlines the climate extreme



indices and evaluation methodology. Results are presented in Section 4, followed by further
discussion in Section 5. Finally, Section 6 offers a summary and conclusions.

**2.  Data**
**2.1 ERA5**
ERA5 is a global atmospheric reanalysis dataset developed by ECMWF (Hersbach, et
al. 2020). ERA5 provides hourly estimates of many atmospheric, land, and oceanic climate
variables. The data is on a ~30 km horizontal grid and resolves the atmosphere using 137 levels
from the surface up to a height of 0.01hPa (~80 km).
ERA5 is constructed upon the foundation of the Integrated Forecasting System (IFS)
Cy41r2. This allows ERA5 to benefit from a decade's worth of development in areas such as
model physics, core dynamics, and data assimilation techniques. ERA5 is a significant
advancement over its predecessors (e.g., ERA-Interim) due to its higher spatial and temporal
resolution, improved assimilation techniques, and more sophisticated modelling components.
It provides a detailed and accurate representation of various atmospheric variables, such as
temperature, humidity, wind speed, pressure, and more. The dataset covers the entire globe and
spans from 1940 to the present, making it valuable for various applications in climate research,
meteorology, environmental science, and more.
**2.2 BARRA**
BARRA is a high-resolution regional atmospheric reanalysis dataset developed by the
Australian Bureau of Meteorology, which is available from January 1990 to February 2019 (Su,
et al. 2019). BARRA was constructed based on the Australian Community Climate Earth-
System Simulator (ACCESS) model with assimilation of a wide range of observational data to
create a coherent and consistent representation of past weather and climate conditions. BARRA
covers the Australian continent, New Zealand, part of Asia and some Pacific Islands with a



horizontal resolution of 12 km and 70 vertical levels from the surface up to a height of 80 km.
BARRA specifically focuses on providing detailed information about weather patterns and
atmospheric variables over the Australian region, which provides about 100 parameters at
hourly intervals.
**2.3 AGCD**
The observational data in the study are from the Australian Gridded Climate Dataset
(AGCD, Evans et al. 2020). The daily gridded maximum and minimum temperatures, and
precipitation data has a spatial resolution of $0.05°$ ($\sim$ 5km) and is interpolated from observations
at stations across the Australian continent. Most of those stations are in the more heavily
populated coastal regions with far fewer stations inland and over high elevation areas. For
example, there are very few station observations near the Gibson dissert region in Western
Australia, making the gridded observations unreliable over that region. Thus, in the following
figures that region has been masked and not considered for evaluation. Since observations and
reanalyses are not at the same spatial resolutions, we aggregate the observations to the native
grid of ERA5 and BARRA respectively for comparison, including the performance of
statistical significance tests. For comparison purpose, we also interpolate reanalysis to AGCD
grids using the conservative area weighted re-gridding scheme from the Climate Data
Operators (Schulzweida et al., 2006), which will be shown in the Supplementary Information.
The states and sub-regions in the Australian region we discuss in the following can be found
in Figure S1.

**3. Methodology**
**3.1 ET-SCI**
While extreme climate and weather events are generally multifaceted phenomena, in
this study we evaluate climate extremes based on daily precipitation and temperature as defined



by Expert Team on Sector-specific Climate Indices (ET-SCI; Alexander & Herold, 2015;
Herold and Alexander, 2016). We use the ClimPACT version 2 software to calculate the ET-
SCI indices (https://climpact-sci.org/), focussing on daily precipitation.
Although ClimPACT generates 14 precipitation-related core indices, we select six
(Table 1) based on the following considerations: 1) To capture key aspects of climate extremes,
we include absolute indices such as the maximum 1 day precipitation (Rx1day) and total
precipitation (PRCPTOT), threshold-based indices (e.g., number of heavy rain days, R10mm),
percentile indices (e.g., total annual precipitation from very heavy rain days, R99p), and
duration indices such as the consecutive wet (CWD) and dry days (CDD). 2) to capture
extremes which have an impact on society and infrastructure, such as Rx1day, CDD, and CWD,
which significantly affect agriculture, water resources and the economy (Tabari, 2020; Pei et
al., 2021).
With the above consideration, six precipitation-related indices were calculated on
native reanalysis grids and observation grids. Since the AGCD observations have the highest
resolution, here we mainly show the evaluation on the native grids of the reanalyses (i.e., the
12-km grid of BARRA and 30-km grid of ERA5). The extreme indices calculated from
reanalysis data have also been regrided to the 5-km resolution, which are included in the
supplementary information to demonstrate that our conclusions are insensitive to the choice of
evaluation resolution.
**3.2 Evaluation matrices**
We evaluate BARRA and ERA5 for their performance in capturing climatology,
coefficient of variation (CV), temporal correlation, and trends of six selected precipitation
extreme indices. The CV is a valuable statistical tool representing the ratio of the standard
deviation to the mean, allowing for the comparison of variation between different data series,
even when their means differ significantly. Temporal correlations of climate extremes measure



the similarities between simulations and observations in terms of their inter-annual variabilities,
with larger temporal correlations indicating better performance.
We use bias and domain-averaged absolute bias to quantify spatial differences between
reanalyses and observations. Temporal correlation, coefficient of variation, and trend are used
to quantify temporal similarities between reanalyses and observations. The non-parametric
Mann-Kendall test is used to assess the statistical significance of differences and trends. Biases
are assessed at an annual timescale for all extremes.

### 213 4. Results

### 214 4.1 Mean climate

This section evaluates and compares the annual mean of daily precipitation between
BARRA and ERA5 against AGCD over Australia.

#### 217 4.1.1 Bias and temporal correlation

We first evaluate precipitation simulated by BARRA and ERA5 against observations
(AGCD). The mean annual precipitation from the three datasets and biases in BARRA and
ERA5 compared to AGCD are shown in Figure 1 (and Figure S2 on the observation grid).
Results show that both BARRA and ERA5 simulate the spatial patterns of mean annual
precipitation very well with high rainfall in northern Australian, eastern Australia coast and
western Tasmania and low rainfall inland, albeit with clear biases. Compared to AGCD, both
BARRA and ERA5 underestimate precipitation up to 20% for Eastern Australian coast,
southwest western Australia, and western Tasmania, but overestimate annual precipitation up
to 30% inland (Figure S3). Some clear differences in biases between BARRA and ERA5 can
be observed in central western Australia and northern Queensland where BARRA overestimate
precipitation but ERA5 underestimate it. Domain averaged absolute bias in annual precipitation
is about 0.17mm/day (~12.7%) for BARRA and 0.15 mm/day (~10.5%) for ERA5 (Table 2).


230 The skill of simulated precipitation from BARRA and ERA5 are further demonstrated

231 in the temporal correlations between BARRA/ERA5 and AGCD shown in Figure 2 (and Figure

232 S4 on the observation grid). Temporal correlation of annual precipitation is larger in southeast

233 Australia and northern Tasmania for both BARRA and EAR5, which is above 0.85. This

234 indicates inter-annual variability of precipitation is well captured by BARRA and ERA5. In

235 contrast, temporal correlation is weaker for western inland and northern Australia. ERA5

236 generally has larger temporal correlation when compared with BARRA, especially for northern

237 Australia, where temporal correlation for BARRA is below 0.5. On average, temporal

238 correlation for ERA5 is 0.85, which is large than 0.77 for BARRA (Table 2).

239 **4.1.2 CV (coefficient of variation) and trend**

240 CV of annual precipitation for AGCD and biases between BARRA/ERA5 and AGCD

241 are presented in Figure 3 (and Figure S5 on the observation grid). By its definition, CV helps

242 capture the standard deviation in the dataset relative to its mean. In the observation, CV is

243 generally smaller for coastal regions except for northwest West Australia and Tasmania than

244 inland Australia, where annual rainfall is much smaller than coastal regions. Alternatively,

245 regions with higher annual precipitation generally have smaller CV. Both BARRA and ERA5

246 reasonably capture the main feature of CV in observation. However, clear biases can be

247 observed, especially in BARRA that has more than 50% large positive biases in Northern

248 Australia, up to 20% positive biases for inland, and relatively smaller biases for southeastern

249 Australia, southwest West Australia and Tasmania. In contrast, ERA5 does not have a clear

250 bias pattern and biases are relatively smaller when compared to BARRA.

251 To further investigate the variability evident in observations and BARRA/ERA5

252 simulations, we assess the trends in annual precipitation (Figure 4 and Figure S6 on the

253 observation grid). AGCD shows strong increasing trends over Northern Australia and

254 Northeast Australia coastal regions but decreasing trends over Northern Queensland,



southwestern West Australia and southern Great Dividing Range including Victoria, although
not all trends are significant. Most of inland regions have relatively small trend in annual
precipitation. Both BARRA and ERA5 reproduce the major trend pattern reasonably well,
however, clear biases can be observed over Northern Australia where both BARRA and ERA5
underestimate biases more than 100%.  BARRA overestimated decreasing trend over Northern
Queensland but ERA5 underestimate it (even increasing trend instead).

In summary, evaluation of annual mean precipitation indicates both BARRA and ERA5

possess small biases (~20%) in the spatial precipitation patterns. ERA5 shows stronger
temporal correlations than BARRA, particularly in northern Australia. Overall, ERA5
demonstrates higher accuracy in capturing inter-annual precipitation variability. Both BARRA
and ERA5 captured spatial distribution of coefficient of variation reasonably well but with
large biases (~ 50%). BARRA shows much larger biases than ERA5 especially for Northern
Australia.  Both BARRA and ERA5 roughly reproduce the pattern of trend but with very large
biases (~100%), especially for Northern Australia where both substantially underestimate the
increasing trend.

**4.2  Climate extremes**

This section evaluates the six select precipitation extreme indices (Table 1) from

BARRA and ERA5 over Australia by comparing them against AGCD. Evaluations are
performed primarily using spatial bias maps and temporal correlations. We also assess the
interannual variability and trends in the simulated BARRA and ERA5 indices and compare
these with AGCD to further investigate any discrepancies.
**4.2.1   Bias and temporal correlation**

Annual mean biases in the six precipitation extremes are shown in Figure 5 (and Figure

S8 on the observation grid). For duration-related extremes, there is a clear north-to-south



gradient in AGCD (Figure S7) with longer duration of CDD and CWD in northern Australia
than southern Australia (CWD also has a clear west-to-east gradient in Tasmania), which is
well simulated in BARRA and ERA5 (Figure S7). While the spatial distributions are well
captured, clear biases are evident in them (Figure 5). BARRA generally underestimates CDD
especially for central inland and northwest West Australia where biases are up to 40%. ERA5
also under-estimates CDD for central inland, but in contrast its over-estimates CDD for most
of northwestern Australia, overall ERA5 has smaller absolute bias in CDD (6.9 days) than
BARRA (14.5 days) (Table 2). Both BARRA and ERA5 have similar bias pattern for CWD,
which generally overestimate CWD over most of regions except for southern Australian coast,
southwest West Australia and western Tasmania. The positive biases over Northern Australia
may reach 30%. Overall BARRA has slightly larger biases in CWD (2.3 days) than ERA5 (1.7
days) (Table 2).

Both BARRA and ERA5 also generally match the spatial distribution of heavy rainfall

days and R90p (Figure S7) in AGCD with large values in Northern Australia, eastern seaboard
and Australian Great Dividing Range, and western Tasmania. However, clear biases can be
observed in BARRA and ERA5 for both R10mm and R90p (Figure 5). BARRA and ERA5
have large negative biases in R10mm over Northern Australia, eastern seaboard, southwest
Western Australia and western Tasmania, but biases in central inland and northwest West
Australia are generally small. Overall, domain averaged absolute bias for ERA5 (1.7 days) is
about half of that for BARRA (3.3 days). Both BARRA and ERA5 also have relatively large
negative biases in R90p for most of northern Australia, eastern coasts, southwest West
Australia and western Tasmania but small positive biases inland, especially for BARRA.
Overall averaged absolute bias is 0.78 mm/day for BARRA and 0.44 mm/day for ERA5 (Table

2).



BARRA and ERA5 also reasonably captured the spatial patterns of R99p and Rx1day,
however, quite large biases are in BARRA and ERA5 (Figure 5). BARRA generally
overestimate R99p and Rx1day over northern Australia coasts and along the Great Dividing
Range, in contrast, ERA5 generally underestimate R99p and Rx1day over northern and eastern
coasts, southwest Western Australia and western Tasmania. The domain averaged bias in R99p
is at similar magnitude for BARRA (4.09 mm/day) and ERA5 (3.67 mm/day), however biases
in Rx1day is much larger for BARRA (20.3 mm/day) than ERA5 (7.9 mm/day) (Table 2).
Figure 6 (and Figure S9 on the observation grid) presents the temporal correlations
between BARRA/ERA5 and AGCD for the six precipitation extreme indices. Unlike the strong
temporal correlation between BARRA/ERA5 and AGCD for mean annual precipitation (Figure
2), the temporal correlations for these extreme indices are worse except for R90p (Figure 6).
For extremes like R10mm and R90p, the correlation ranges from reasonably good (above 0.6)
to pretty good (above 0.8) between BARRA/ERA5 and AGCD for most of the domain.
Temporal correlation for CWD and R99p are not as good as R10mm and R99p, but they are
comparatively stronger correlations (0.5-0.6) than CWD and Rx1day (~0.5 and less) over most
of the domain. Compared to BARRA, ERA5 has slightly stronger temporal correlations for
those extremes (Table 2).

**4.2.2  CV (coefficient of variation) and trend**
The observed and simulated CV of precipitation extremes and biases in their CV for
BARRA and ERA5 are shown in Figure S10 and Figure 7 (and Figure S11 on the observation
grid), respectively. Generally, both BARRA and ERA5 have similar CV bias patterns and
magnitude for CDD, CWD and R10mm. In contrast, BARRA is quite different from ERA5 for
other three extremes. BARRA substantially under-estimated CV of R90p over most on inland
regions but ERA5 has much smaller negative biases, even small positive biases, although both



have small biases in CV of R90p along most coastal regions and Tasmania. BARRA
systematically overestimate CVs of R99p and Rx1day over northern Australia but ERA5 has
relatively small biases for them. Overall, BARRA has more than twice as much as CV biases
in ERA5 for R90p, R99p and Rx1day (Table 2).

Trends of each of the precipitation extreme indices for the three datasets and biases in

trend for BARRA and ERA5 are shown in Figure S12 and Figure 8 (and Figure S13 on the
observation grid), respectively. Generally, both BARRA and ERA5 simulate the main pattern
of trends for those extremes but with large biases. BARRA and ERA5 simulated CDD trend
well for southern Australia but BARRA generally under-estimated trend in CDD over inland
Australia and overestimate trend in northwest Australia. ERA5 only has large positive trend
biases in northern central Australia. The overall domain averaged biases are similar between
BARRA (0.584) and ERA5 (0.566). Both BARRA and ERA5 have small biases in CWD in
central and southern Australia but similar biases pattern in Northern Australia. They also have
similar overall biases in CWD (0.064 for BARRA and 0.060 for ERA5). Both BARRA and
ERA5 under-estimated increasing trend in R10mm in northern Australia, but BARRA
overestimate trend in most of southeast Australia. In contrast, ERA5 under-estimate trend over
there. Overall, ERA5 has slightly larger biases (0.094) than BARRA (0.085). Like R10mm,
both BARRA and ERA5 also underestimate trend of R90p in most of northern Australia but
have small biases in central and southern Australia. They have almost the same overall biases
in R90p. BARRA/ERA5 has similar biases patterns for R99p and rx1day but biases for rx1days
are much larger. Both BARRA and ERA5 have large biases in R99p and Rx1day but biases in
BARRA are generally larger than ERA5.

In summary, both BARRA and ERA5 reproduce spatial patterns of extremes well but

display biases. ERA5 underestimates CDD and certain heavy rainfall events, while BARRA
tends to overestimate these extremes. Both reanalyses show discrepancies in various



precipitation indices across different regions, with BARRA generally displaying larger biases
compared to ERA5. Temporal correlations between BARRA/ERA5 and observations for
extreme precipitation indices are weaker than those for mean annual precipitation, except for a
few indices where ERA5 demonstrates slightly stronger correlations compared to BARRA.
Both BARRA and ERA5 align in CV patterns and biases for certain extremes but differ notably
in others. BARRA significantly underestimates very heavy precipitation variability over inland
regions, while ERA5 presents smaller biases or even positive biases in these areas. Additionally,
BARRA tends to overestimate extreme precipitation variability in Northern Australia
compared to ERA5. Overall, BARRA shows more than double the biases in variability
compared to ERA5 for specific extreme precipitation indices. Both reanalyses generally
simulate the main trend patterns but exhibit considerable biases. BARRA underestimates or
overestimates trends in certain regions and indices, while ERA5 demonstrates different biases,
including smaller biases overall compared to BARRA across these precipitation extremes.

**5. Discussion**
In this study, we assessed the performance of BARRA and ERA5 in simulating mean
precipitation and six selected precipitation extremes. While most previous evaluations have
focused on the climatology of precipitation and its extremes, only a few studies have included
the coefficient of variation (CV) (Teng et al., 2024). Our evaluation encompassed annual
climatology, along with temporal correlation, CV, and trend analysis, providing a
comprehensive assessment of the performance of these two reanalysis datasets.
The results indicate that both BARRA and ERA5 demonstrate reasonable skill in
simulating mean precipitation and certain precipitation extremes. However, they encounter
challenges in accurately reproducing temporal correlation, CV, and trends for certain extreme
events, highlighting significant uncertainties in their representation of extremes.



While acknowledging the capabilities of these reanalysis datasets, our study also
identifies specific limitations and suggests potential directions for future research. A crucial
consideration in model evaluation is the accuracy of observational data, which substantially
influences evaluation outcomes. In this study, we used the AGCD dataset as the observational
benchmark, which is based on interpolating data from in-situ stations (Evans et al., 2020).
However, the AGCD dataset presents several limitations: 1) Spatial coverage: Sparse station
coverage in northwest and central Australia, and limited observations in high-elevation areas,
result in a concentration of stations in southeastern Australia, southwestern Western Australia,
and eastern Tasmania. The arid interior is notably underrepresented. 2) Data completeness and
homogeneity: Incomplete and inhomogeneous observations due to missing data, changes in
observational techniques, or station relocations can affect the consistency of the dataset. 3)
Interpolation uncertainties: The interpolation method used in AGCD (splining), instead of the
ordinary kriging method used in its predecessor (AWAP), introduces uncertainties, particularly
in areas with sparse data coverage for extreme events like heavy rainfall.
These observational uncertainties may contribute to biases in the evaluation results. In
particular, the limited number of monitoring sites over the Great Dividing Range and inland
areas introduces significant uncertainties in estimated observed precipitation for these regions.
Independent studies, such as Chubb et al. (2016), found that daily precipitation is
underestimated by at least 15% in some areas, which could suggest similar underestimation in
BARRA and ERA5 for these regions. Similarly, the sparse gauge network in northwestern
inland areas might miss localized extreme precipitation events.
Our analysis focused on six ET-SCI-defined precipitation extreme indices, widely used
in various evaluation studies (Nishant et al., 2020; Ji et al., 2024). However, recognizing the
need for region-specific indices, we suggest future studies extend the analysis to incorporate
additional extreme indices tailored to specific regions and applications.





Our findings emphasize that while both BARRA and ERA5 are competent in simulating
the climatology of mean climate, temporal correlation, and CV, challenges remain in accurately
capturing trends, particularly for certain extremes. Notably, ERA5 shows better overall
performance compared to BARRA. Although higher resolution often correlates with better
performance, recent studies have shown that increasing resolution alone does not always
guarantee improvements (Nishant et al., 2022). Considering the critical role of driving data,
model physics, and data assimilation, it may be valuable to update BARRA using the latest
ERA5 data along with improved model physics and data assimilation techniques to enhance its
performance.
In this study, we evaluated ERA5 and BARRA on both their native resolutions and a
common resolution (5 km) to match AGCD. The results showed that the evaluations were
consistent across native and common resolutions, suggesting that the performance assessments
were not highly sensitive to changes in resolution.

**6. Summary and Conclusion**
Reanalysis datasets play a crucial role in climate research, weather analysis, and various
scientific investigations. Their ability to provide a consistent and comprehensive representation
of historical weather and climate conditions makes them invaluable. These datasets are
particularly essential for studying long-term climate trends, understanding atmospheric
phenomena, and validating climate models.
In this study, we evaluate BARRA and ERA5 for their capabilities to simulate mean
precipitation and six selected precipitation extremes for their climatology, temporal correlation,
coefficient of variation (CV) and trend to quantify their overall performance. We evaluated
BARRA and ERA5 at their native resolutions, as well as at a common resolution (i.e., the



observation resolution). Both analyses yielded consistent results, indicating that the evaluation
is not sensitive to the remapping process.
The assessment of annual mean precipitation reveals that both BARRA and ERA5
adeptly reproduce the spatial precipitation patterns, exhibiting minor biases of around 20%.
Particularly, ERA5 showcases stronger temporal correlations compared to BARRA, especially
evident in northern Australia. ERA5, overall, demonstrates superior accuracy in capturing
inter-annual precipitation variability. However, both models depict the spatial distribution of
the coefficient of variation reasonably well but with larger biases, roughly around 50%.
Particularly, BARRA displays significantly higher biases, especially in Northern Australia.
Regarding the replication of trend patterns, both models exhibit substantial biases,
reaching approximately 100%. This is especially notable in Northern Australia, where they
both notably underestimate the increasing trend. Furthermore, while both BARRA and ERA5
possess about the right spatial patterns of extremes, biases are evident. ERA5 tends to
underestimate consecutive dry days (CDD) and certain heavy rainfall events, while BARRA
tends to overestimate these extremes. Discrepancies in various precipitation indices across
regions are apparent, with BARRA generally displaying larger biases compared to ERA5.
When examining temporal correlations for extreme precipitation indices compared to
mean annual precipitation, both BARRA and ERA5 show weaker correlations, except for a
few indices where ERA5 slightly outperforms BARRA. While both models align in coefficient
of variation patterns and biases for certain extremes, notable differences arise in others.
BARRA notably underestimates very heavy precipitation variability over inland regions,
whereas ERA5 presents smaller biases or even positive biases in these areas. Moreover,
BARRA tends to overestimate extreme precipitation variability in Northern Australia
compared to ERA5. Specifically, BARRA showcases more than double the biases in variability
compared to ERA5 for specific extreme precipitation indices.



In terms of trend patterns, both models generally replicate the observed trends but
exhibit considerable biases. BARRA shows both underestimations and overestimations in
certain regions and indices, while ERA5 displays different biases, including overall smaller
biases compared to BARRA across these precipitation extremes.
In summary, our findings suggest that both ERA5 and BARRA are reliable for
climatological analyses, including mean precipitation and precipitation extremes, and can be
confidently used by end-users for such purposes. However, as discussed in the introduction,
caution is advised when using these datasets for variability and trend analyses, particularly for
specific extreme events like Rx1day. The performance of these reanalyses is regionally
dependent, and this should be considered when using them as observational references for
evaluating other model simulations. Additionally, the biases in the variability and trends of
climate extremes present in both datasets must be carefully accounted for when comparing
them with other data sources.


**Data Availability**

Details about AGCD are available at the Australian Bureau of Meteorology website
(http://www.bom.gov.au/metadata/catalogue/19115/ANZCW0503900567, (accessed on)).
The dataset is available on the NCI (National Computational Infrastructure) server in project
zv2. Detail on how to access the data can be found at http://climate-
cms.wikis.unsw.edu.au/AGCD, (accessed on). ERA5 data is available on the NCI in Project
rt52. BARRA data is available on the NCI in project cj37.

**Author Contributions**

KKWC and FJ conceptualized and implemented the research. KKWC, FJ and NN performed
the data analysis and prepared the figures. FJ prepared the draft manuscript. All authors
contributed to the discussion of results, editing and finalization of the manuscript.

**Competing Interests**

The authors declare that they have no conflict of interest.

**Acknowledgments**



KKWC acknowledges support from The Startup Foundation for Introducing Talent of the
Nanjing University of Information Science and Technology. The modelling work was
undertaken on the National Computational Infrastructure (NCI) high performance computers
in Canberra, Australia, which is supported by the Australian Commonwealth Government.



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





Table 1 List of ET-SCI indices evaluated in this study.

| Index | Definition | Units | Timescale | Sectors |
|---|---|---|---|---|
| PRCPTOT | Total wet-day precipitation (Sum of daily precipitation >= 1.0 mm) | mm | Annual/Monthly | Agriculture and food security, water, water resources and food security, forestry/GHGs |
| CDD | Consecutive dry days (Maximum number of consecutive dry days (when precipitation < 1.0 mm)) | days | Annual | Health, agriculture and food security, water resources and food security, disaster risk reduction, forestry/GHGs |
| CWD | Consecutive wet days (Maximum annual number of consecutive wet days (when precipitation >= 1.0 mm)) | days | Annual | Coasts, agriculture, transport operations |
| R10mm | Days when precipitation is at least 10mm | days | Annual/Monthly | Coasts |
| R90p | Total annual precipitation from very heavy precipitation days (Annual sum of daily precipitation > 90th percentile) | mm | Annual | Coasts, transport operations |
| R99p | Total annual precipitation from very heavy precipitation days (Annual sum of daily precipitation > 99th percentile) | mm | Annual | Coasts, transport operations |
| Rx1Day | Amount of precipitation from very wet days (Maximum 1-day precipitation) | mm | Annual/Monthly | Agriculture and food security, water, coasts, disaster risk reduction, forestry/GHGs |

Table 2 Domain-averaged absolute biases and temporal correlation between BARRA/ERA5 and AGCD for annual precipitation and precipitation extremes

| Indices | Absolute biases in annual mean | | Temporal correlation | | Absolute biases in CV | | Absolute biases in trend | |
|---|---|---|---|---|---|---|---|---|
| | BARRA | ERA5 | BARRA | ERA5 | BARRA | ERA5 | BARRA | ERA5 |
| Annual pr | 0.169 | 0.149 | 0.771 | 0.854 | 0.063 | 0.037 | 0.008 | 0.007 |
| CDD | 14.543 | 6.913 | 0.578 | 0.650 | 0.050 | 0.045 | 0.584 | 0.566 |
| CWD | 2.346 | 1.714 | 0.446 | 0.527 | 0.061 | 0.059 | 0.064 | 0.060 |
| R10mm | 3.265 | 1.700 | 0.688 | 0.761 | 0.081 | 0.053 | 0.085 | 0.094 |
| R90p | 0.777 | 0.439 | 0.761 | 0.827 | 0.211 | 0.082 | 0.023 | 0.023 |
| R99p | 4.093 | 3.668 | 0.562 | 0.625 | 0.121 | 0.060 | 0.206 | 0.162 |
| Rx1day | 20.333 | 7.916 | 0.380 | 0.486 | 0.219 | 0.107 | 0.848 | 0.542 |

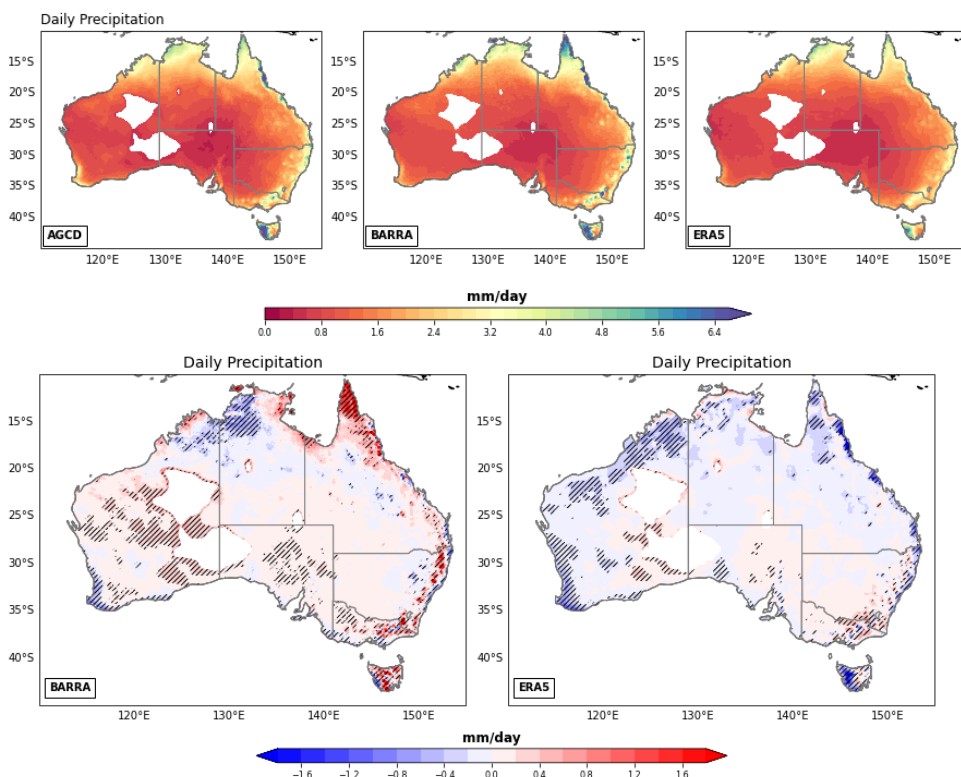

Figure 1 Annual mean precipitation of AGCD, BARRA and ERA5 (upper panels) and annual mean biases between BARRA/ERA5 and AGCD (lower panels). The regions with low density of station observations in AGCD has been masked and not considered in all subsequent evaluation. Unit: mm/day. Stippling indicates areas with biases that are statistically significant at 95% confidence level.





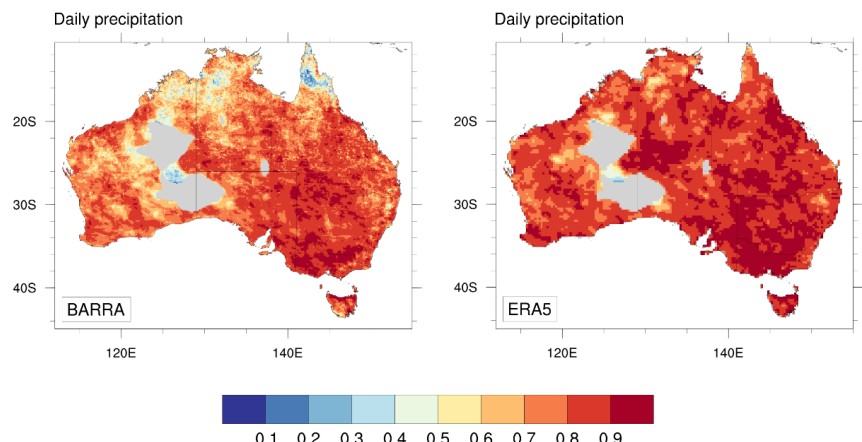

Figure 2 Temporal correlation coefficient of annual precipitation between BARRA/ERA5 and AGCD.

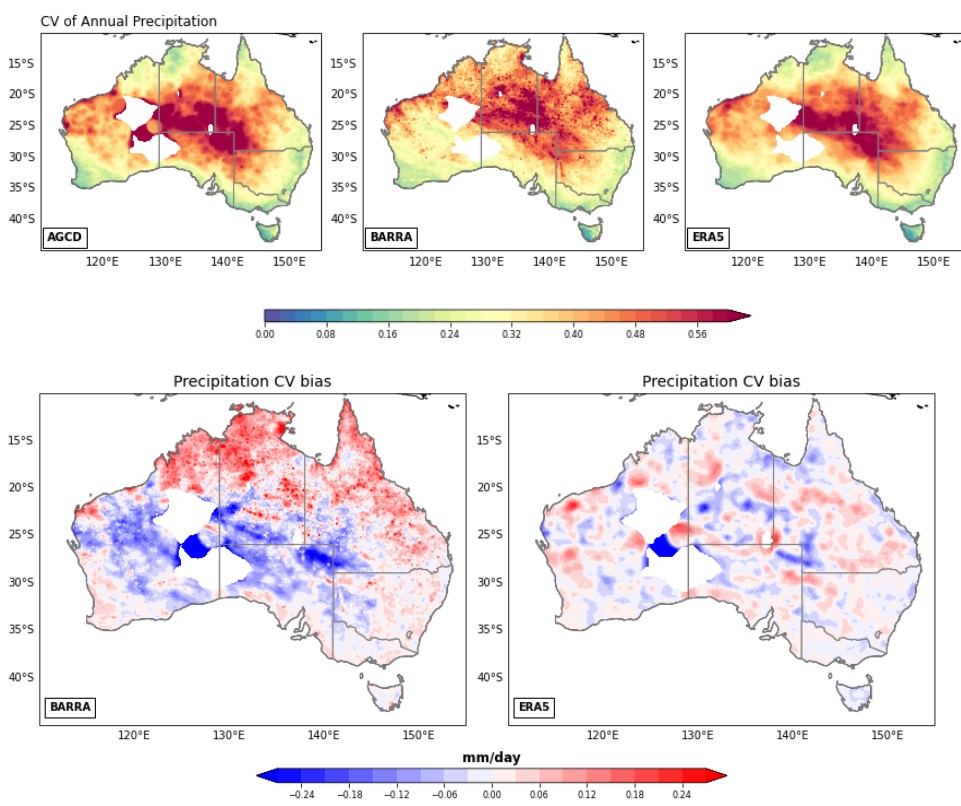

Figure 3 CV of annual precipitation for AGCD, BARRA and ERA5 (upper panels) and biases
in CV between BARRA/ERA5 and AGCD (lower panels).



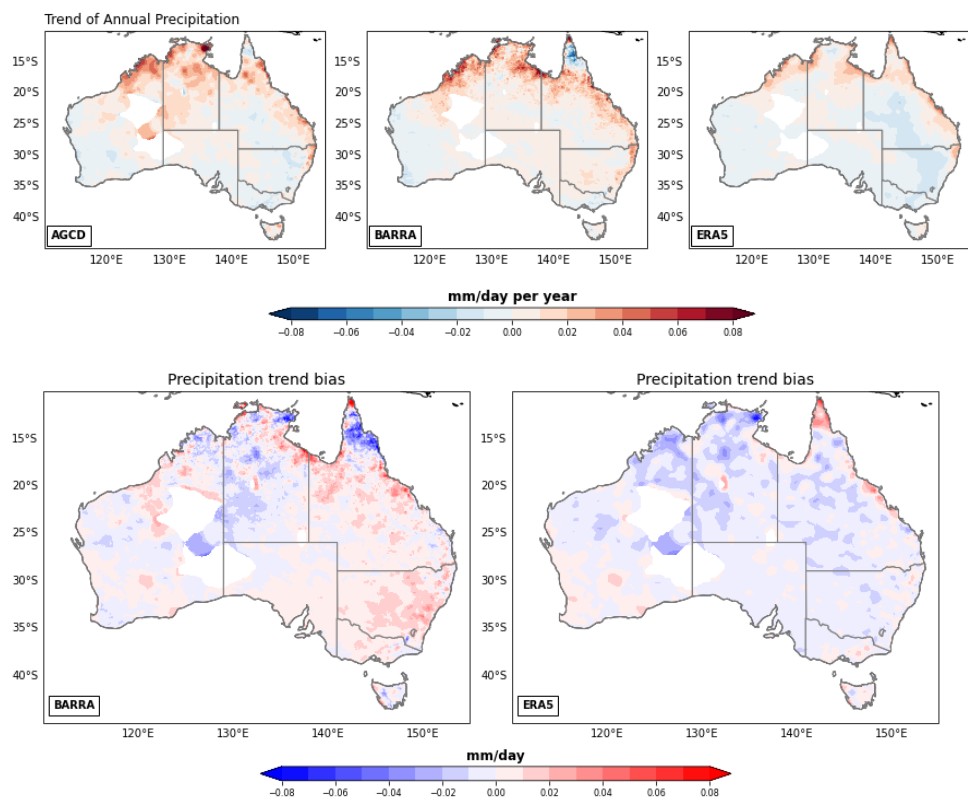

Figure 4 Trend of annual precipitation for AGCD, BARRA and ERA5 (upper panels) and biases in trend between BARRA/EAR5 and AGCD (lower panels).

Figure 5 Biases in CDD, CWD, R10mm, R90p, R99p and Rx1Day in BARRA (left column) and ERA5 (right column). Stippling indicates areas with biases that are statistically significant at 95% confidence level.

Figure 5 (continued).





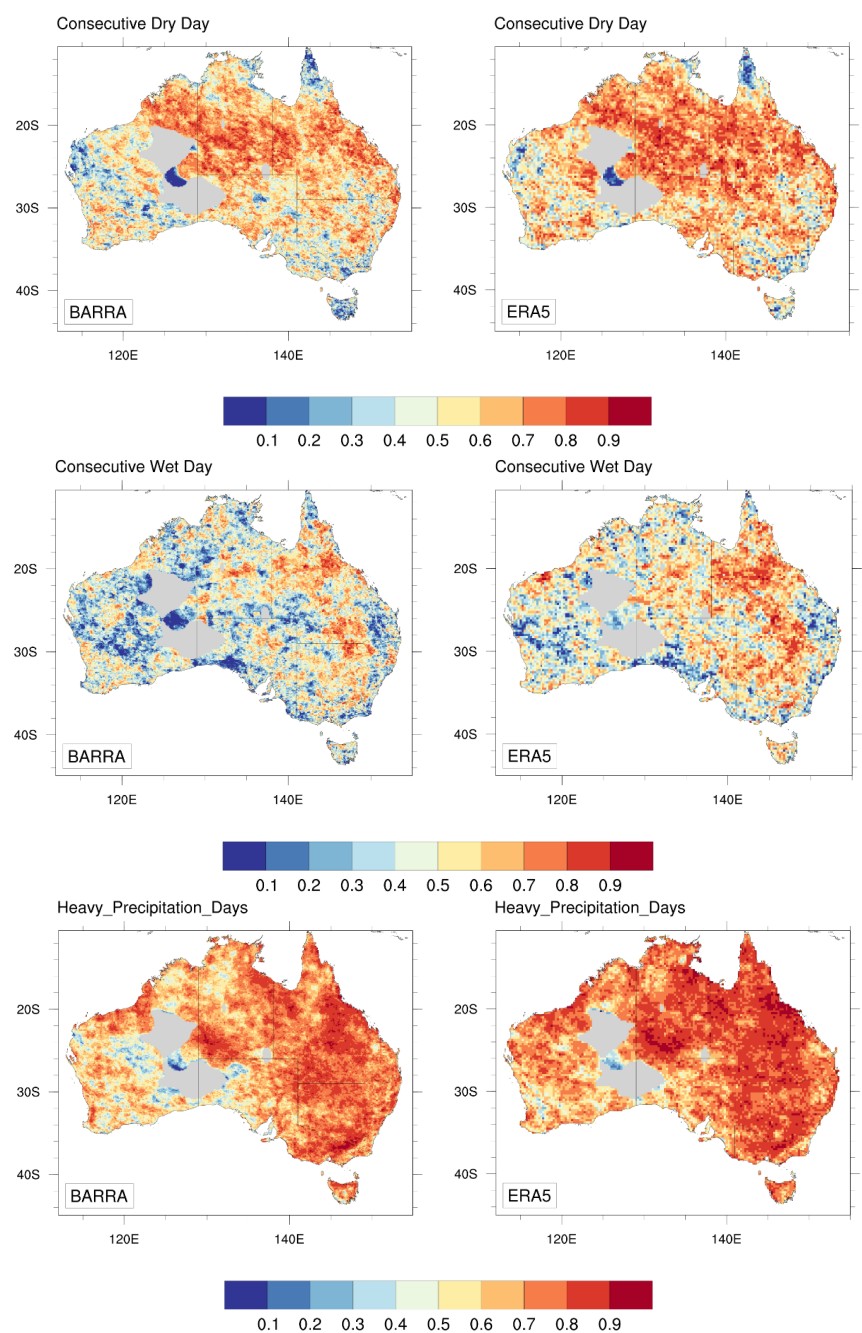

Figure 6 Temporal correlation of CDD, CWD, R10mm, R90p, R99p and Rx1Day between
BARRA and AGCD (left column) and between ERA5 and AGCD (right column).



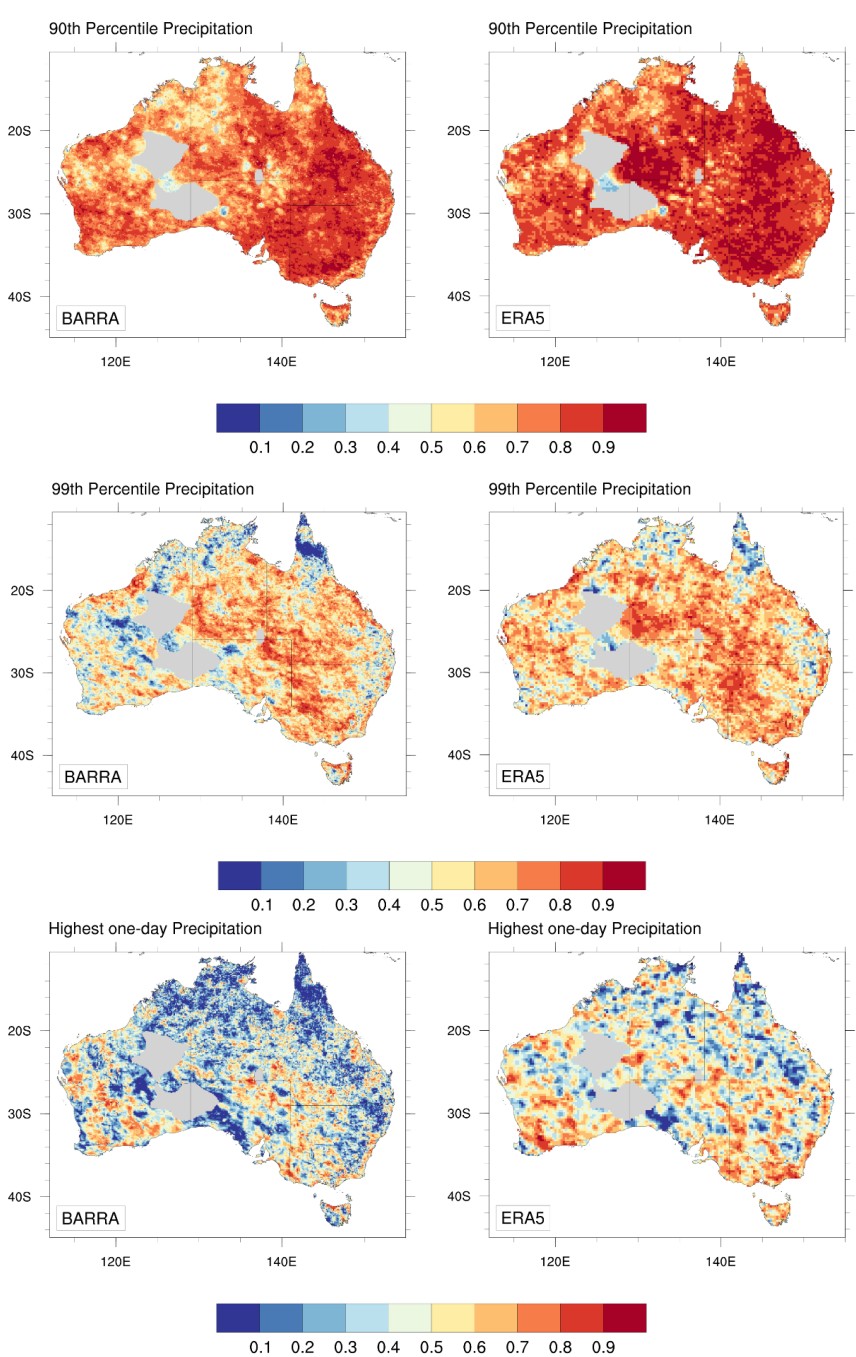

Figure 6 (continued).



Figure 7 Biases in CV of CDD, CWD, R10mm, R90p, R99p and Rx1Day for BARRA (left
column) and ERA5 (right column) relative to AGCD.





Figure 7 (continued).



Figure 8 Biases in trends of CDD, CWD, R10mm, R90p, R99p and Rx1Day for BARRA (left
column) and ERA5 (right column) relative to AGCD.



Figure 8 (continued).