# Peer review of "Comparison of BARRA and ERA5 in Replicating Mean and Extreme Precipitation over Australia"

_Hydrology and Earth System Sciences, 2024_

## Author Comment (AC1)

**Responses to RC1 of "Comparison of BARRA and ERA5 in Replicating Mean and Extreme Precipitation over Australia" by Cheung et al."**

The authors have evaluated BARRA and ERA5 reanalysis data against observed precipitation across Australia, following a comprehensive literature review of previous evaluations of ERA5/ERA Interim and BARRA. Unlike earlier assessments of BARRA that primarily focused on precipitation climatology, this study emphasizes precipitation extremes, temporal correlation, and long-term trends. The evaluation provides valuable guidance for users regarding data analysis and model evaluation based on BARRA data. The manuscript concludes that BARRA exhibits a larger overall bias than ERA5 concerning precipitation extremes. However, the authors do not explain the potential sources of this bias. More analysis or discussions are necessary before the manuscript can be considered for publication in HESS. Detailed comments are as follows:

Responses:
We appreciate the comment from RC1 that our study would provide valuable guidance for users applying BARRA data. We accept your suggestion to increase discussion on the potential sources of biases in BARRA. Following your comments in the following, we will add more analysis and discussion to enhance our manuscript in this aspect, as detailed in responses to the specific comments.

Major Comments:

1. Clarification of BARRA Data: Please provide additional information about the BARRA dataset. For instance, is ACCESS, used to construct BARRA, a regional climate model? What large-scale forcing data was used to drive the regional climate model? What observational data were assimilated into the BARRA dataset? Specifically, was observational precipitation included in the assimilation process? This information is crucial for understanding the results presented in the manuscript.

Responses: The following discussion on BARRA's background information has been added to the revised manuscript.

The Bureau of Meteorology (BoM)'s ACCESS model, which was applied to generate BARRA, originated from the UKMO's Unified Model (UM), which can be configured in global mode or regional mode. For regional simulations, the global version of ACCESS becomes ACCESS-R. ACCESS-R was initialized by ERA-Interim reanalysis data, which also provides boundary conditions during simulation. A series of observations have been assimilated into BARRA, including land and ship (buoy) synoptic observations, upper-air observations from radiosondes and wind profilers, satellite derived radiances and winds (Su et al. 2019). However, no precipitation observations were directly assimilated.

2. Evaluation of Precipitation Extremes: Section 4.1 evaluates the annual mean precipitation derived from BARRA and ERA5, which is partly related to precipitation extremes. It would be desirable to further assess precipitation on a day-to-day basis, such as the correlation and

variance of daily precipitation. From a probability density distribution perspective, both the mean and variance influence extremes. To what extent is the bias in precipitation extremes related to mean and variance biases?

Responses:
Thank you for this comment. In our study, the mean bias and CV do measure the accuracy of mean and standard deviation in the density distribution of precipitation on interannual timescale. We agree that it is highly desirable to examine day-to-day variability of precipitation based on our evaluation measures, a timescale of which influences from transient synoptic to mesoscale weather systems are important. However, it is quite outside the original scope of our study. Out of curiosity, we have examined mean precipitation, standard deviation and CV based on daily data from AGCD, BARRA and ERA5, as shown in the following figures. It can be seen that on daily timescale their patterns of standard deviation and CV deviate from each other quite substantially. We will put such a daily evaluation as a follow-up study of the current one.

[Figure]

[Figure]

3. Investigation of Reanalysis Biases: The manuscript lacks an investigation or discussion regarding the sources of reanalysis biases. BARRA was constructed based on the ACCESS model with the assimilation of observational data. It appears that the regional climate model (RCM) used to construct BARRA was driven by ERA-Interim. Do ERA-Interim and BARRA exhibit similar biases in precipitation extremes? To what extent does the BARRA inherit biases from ERA-Interim? What role does the parameterization scheme of the RCM play in precipitation biases?

Responses:
To further our response to point #1, the RCM used in BARRA (i.e., ACCESS-R) was driven by ERA-Interim reanalysis. Detailed comparison between the biases in BARRA and ERA-Interim, when evaluated against BoM's in-situ observations, have been performed in Su et al. (2019). In general, BARRA shows better agreement with point-scale observations of 2-m temperature, 10-m wind speed and surface pressure. Some biases of BARRA are indeed inherited from ERA-Interim, such as negative biases in strong wind speed. Monthly time series from BARRA and ERA-Interim (e.g., for maximum and minimum temperature and precipitation) during the evaluation period in Su et al. (2019) (2003–2016) also show similarities. Both BARRA and ERA-Interim did not assimilate observed precipitation directly, and in the 12-km grid BARRA did apply convection parameterization (the mass flux convection scheme of Gregory and Rowntree 1990). BARRA have better similarities with AWAP, the gridded observational dataset of BoM's rain gauges, than ERA-Interim, especially in terms of frequency statistics for heavy rain events and annual mean.

Other Comments:

L42 and elsewhere: Since ERA5 and BARRA are not solely model outputs but also results of extensive observational data assimilation, the statement "Both 'models' reproduce spatial patterns of mean precipitation well" is misleading. The authors may consider replacing "model" with "dataset."

Responses: We have replaced "model" by "dataset" in the revised manuscript.

L232-234 and Figure 2: To my understanding, Figure 2 illustrates the correlation coefficient of annual precipitation between reanalysis and AGCD over the period from 1990 to 2019. This

assesses the ability of reanalysis data to reproduce interannual variation of annual precipitation. How well do the reanalysis datasets reproduce observed day-to-day precipitation variability in various seasons?

Responses: Please see our response to major comments #2.

L259: What is meant by "underestimate biases"?

Responses: Thanks for picking this up. "underestimate biases" was incorrect - we meant both BARRA and ERA5 underestimate the trend. This has been corrected in the revised manuscript.

Figures 3, 4, 6, 7, 8: Please also indicate the significance of bias, trend, or correlation in these figures.

Responses: Thanks for this suggestion. Unfortunately, we have only retained the mean values of trend and CV during the evaluation period but not the entire sample, and thus not able to compute significance of trend and their biases.

Figure 7: Please also evaluate the coefficient of variation (CV) of day-to-day precipitation in different seasons.

Responses: Please see our responses to major comment #2.

L279-281, Figure S7: Both consecutive dry days (CDD) and consecutive wet days (CWD) exhibit longer durations in northern Australia compared to the southern regions. Why is this the case? CDD and CWD usually exhibit opposite changes. Are the CDD and CWD values illustrated in Figure S7 the maximum values observed over one year? The authors may want to evaluate climate extreme indices in different seasons, as northern Australia is influenced by the Asian-Australian monsoon, which presents a distinct annual cycle in precipitation. The climate extreme indices, such as CDD and CWD, can vary significantly across seasons.

Responses: Northern Australia is monsoonal, with very strongly delineated wet and dry seasons. In general, wet seasons (approximately Nov-Mar) are intensely wet, while it seldom rains in the dry season (Apr-Oct). This is why CDD and CWD can both be longer than in southern Australia, which exhibits something like a mediterranean climate. The CDD and CWD values shown in Figure S7 are averaged over the 29-year period. We have examined their values (based on AGCD observations) in four seasons respectively. From the figure below, the clear seasonal variation of the two indices is evident. The highest CDD values at northern Australia occur during spring (SON), which is close to the annual mean pattern. CWD has highest values during autumn (MAM) also at northern Australia and that has been shown in the annual mean as well.

[Figure]

L409-412: Why does BARRA generally perform worse than ERA5? BARRA was produced using a limited-area meteorological forecast model driven by ERA-Interim (Su et al., 2019, GMD). How does BARRA's performance compare with its large-scale forcing data, ERA-Interim, in terms of precipitation? Does BARRA inherit biases from ERA-Interim?

Responses: In our response to major comments #1 and #3 we have briefly summarized the findings in Su et al. (2019) in evaluating BARRA and comparing with the driving reanalysis ERA-Interim. The key points are that BARRA generally agree better with station observations (for surface temperature, winds and precipitation) than ERA-Interim. Indeed, bias patterns and interannual trends in BARRA can be seen to have inherited from ERA-Interim. In this study, on the other hand, we compare BARRA versus ERA5. Thus, relative biases between the two datasets may be related to improvements (in resolution, data assimilation and process representation) of ERA5 over ERA-Interim. Impacts from these improvements are highly complex and inter-related. We will extend our scope in a further study to investigate factors behind differences between BARRA and ERA5. Since ERA5 is currently the most popular dataset for climate evaluation studies, our work has clarified the added-value and inadequacy of BARRA in terms of climate extremes.

---

## Author Comment (AC2)

**Responses to RC2 of "Comparison of BARRA and ERA5 in Replicating Mean and Extreme Precipitation over Australia" by Cheung et al."**

**General comments**

This paper compares a local reanalysis (BARRA) with the global reanalysis of ERA5, and assesses their performance relative to observations in Australia (AGCD). In addition to analysing long-term means, they also assess the ability of BARRA and ERA5 to accurately detect climate extremes. The authors use climate indices from CLIMPACT (a project supported by the World Meteorological Organization): the total precipitation (PCRCPTOT), the maximum 1-day precipitation (Rx1day), the number of heavy rain days (R10mm), the total annual precipitation from very heavy rain days (R99p and R90p), the number of consecutive wet days (CWD) and the cumber of consecutive dry days (CDD).

A good overview of previous work is provided in the introduction, both globally and for Australia and New Zealand. The summary of literature shows that no systematic comparison of ERA5 and BARRA has been conducted for precipitation over Australia. The gap to bridge is clear and the structure of the paper is clearly introduced. Maps and tables are provided to support interpretations and assessment of the quality of the results. Assessing the ability of the reanalysis datasets to reproduce climate extremes is valuable, even if using many climate indices can at times be confusing to the reader (see specific comments). The authors conclude that both ERA5 and BARRA reproduce long-term precipitation patterns with minor biases. However, if climate extremes are correctly evaluated spatially, their temporality show discrepancies compared to the observations of AGCD.

The abstract summarises well the datasets, methods and conclusions. The supplementary material is plenty, but all figures and tables are cited in the main text. The article is structured and well-written. In general, this paper addresses the issue of the quality of two reanalysis datasets (ERA5 and BARRA) for Australia. Given the societal and scientific importance of accurate detection and prediction of heavy rainfall, dry spells and climatic means, knowing the quality of reanalysis datasets over a given territory is a major issue that is thoroughly explored here.

Response: We appreciate that the reviewer thought that our manuscript's structure is clear and we have filled in a gap in knowledge regarding the societal and scientific importance of applying reanalysis datasets to studying hydrological issues. Thank you for the extensive specific comments provided to us for improving our manuscript. We have provided point-to-point responses in the following.

**Specific comments**

*Introduction*

1. Many datasets are presented in the introduction, but references are not added with the first reference (L83: ERA5-Interim is mentioned, but not presented; L98: MERRA-2 is mentioned without reference; L100 JRA55 is mentioned without reference either). References are added later, but it would be clearer to add them right away.

Response: The following references have been added to first mentioning of ERA-Interim (Dee et al. 2011, QJRMS), MERRA-2 (Gelaro et al. 2017, BAMS) and JRA55 (Kobayashi et al. 2015, JMSJ) in the revised manuscript.

Dee, D., et. Al.: The ERA-Interim reanalysis: configuration and performance of the data assimilation system. Quart. J. Roy. Meteor. Soc., 137, 553–597, https://doi.org/10.1002/QJ.828, 2011.

Kobayashi, S., et al.: The JRA-55 reanalysis: General specifications and basic characteristics. J. Meteor. Soc. Japan, 93, 5–48, https://doi.org/10.2151/jmsj.2015-001, 2015

Gelaro, R., et al.: The Modern-Era Retrospective Analysis for Research and Applications, Version 2 (MERRA-2). Bull. Amer. Meteor. Soc., 30. 5419–5454, https://doi.org/10.1175/JCLI-D-16-0758.1, 2017.

2. In addition, in the lines L65-L78, the authors talk about reanalysis datasets but only introduce ERA5 (add reference). This makes sense as this is the global reanalysis product that is going to be used in the article, but it would be clearer either to name the other datasets used in the cited articles, either to save the introduction of ERA5 to the next paragraph (L79-91), that specifically gives examples of how this dataset has been used.

Response: We have mentioned the other datasets in the cited articles in paragraph L65-L78, before specifically discussing ERA5 in the following paragraph (L79–91). The revised paragraph is:

"These datasets are invaluable for climate studies, weather analysis and model validation as they provide a uniform representation of historical climate conditions. For instance, Quagraine et al. (2020) used five global reanalysis datasets (European Centre for Medium-Range Weather Forecasts Reanalysis ERA-Interim, Dee et al. 2011; ERA5, Herbach et al. 2020; JRA-55, Kobayashi et al. 2015; MERRA2, Gelaro et al. 2017; and NCEP-R2, Kanamitsu et al. 2002) to investigate the variability of West African summer monsoon precipitation, showing all datasets could represent the average rainfall patterns and seasonal cycle. Dai et al. (2023) utilized ERA5 data to estimate rainfall erosivity on the Chinese Loess Plateau, finding rainfall erosivity derived from ERA5 was highly consistent with those derived from the meteorological stations. Cheung et al. (2023) employed ERA5 to evaluate storm conditions in regional climate simulations, demonstrating regional climate models can capture climatology of measurements of storm severity over land including their spatial

patterns and seasonality. Numerous studies have used reanalysis datasets as inputs for regional climate models (RCMs) to evaluate the models' capability in replicating observed climatic patterns (Solman et al., 2013; Ji et al., 2016; Fita et al., 2016, Di Virgilio et al., 2019; Capecchi et al., 2023; Di Virgilio et al., 2024; Ji et al., 2024)."

*Data*

1. Overall, the reader does not know which time period is used for the computation of the climate indices. Are they calculated for the total period of availability of each of the datasets (which would make the comparison spurious), or is a reference period chosen?

Response: We are sorry that we did not mention the analysis time period, which was 1990–2018 applied to all three datasets. This has been added to section 3 on methodology in the revised manuscript. The last paragraph (L192–198) in that section is revised to:

"With the above consideration, seven precipitation-related indices were calculated on native reanalysis grids and observation grids. While the availability of AGCD and ERA5 starts much earlier, the analysis period is 1990–2018, which is the duration of BARRA. Since the AGCD observations have the highest resolution, here we mainly show the evaluation on the native grids of the reanalyses (i.e., the 12-km grid of BARRA and 30-km grid of ERA5). The extreme indices calculated from reanalysis data have also been regridded to the 5-km resolution using bilinear interpolation, which are included in the supplementary information to demonstrate that our conclusions are insensitive to the choice of evaluation resolution."

2. Some information is missing in the presentation of the AGCD dataset, notably the period covered by the observations. ERA5 reanalysis covers a period from 1940 to today, while BARRA covers the period 1990-2019. Which period cover the AGCD data used in the article? In addition, the re-gridding scheme is mentioned but not presented in the Supplementary Information. Finally, the given reference does not allow the reader to access the observation data (the cited report)

Response: AGCD is a long dataset that begins from 1900. As mentioned to our response to the last question, we applied the common period of 1990–2018 for comparison between AGCD, BARRA and ERA5. For re-gridding scheme, note that we presented results from the respective native grids from the datasets in the main text. When we regridded BARRA and ERA5 to the AGCD grid, which has been presented in the Supplementary Information, we applied bilinear interpolation. AGCD is an open dataset that is available from the Bureau of Meteorology Australia. We have emphasized in the data availability section of revised manuscript that the dataset can be provided by the authors upon request.

*Methodology*

*ET-SCI climate indices from CLIMPACT*

1. The authors always mention 6 climate indices, but 7 of them are listed in Table 1: PCRPTOT, R1xday, R10mm, R99p, R90p, CDD and CWD. I assume that PCRPTOT is not included in the "climate extreme indices", but then it has to be presented independently. In addition, R90p is not presented in the paragraph L183-191, even though it is used later.

Responses: PCRPTOT is actually one of the ET-SCI indices, so "6 climate indices" was a typo, which has been corrected to "7 climate indices". R90p has also been added back to the paragraph L183–191 (also in response to one of your technical comments):

"Although ClimPACT generates 14 precipitation-related core indices, we select seven (Table 1) based on the following considerations: 1) To capture key aspects of climate extremes, and 2) to capture extremes which have impacts on society and infrastructure such as agriculture, water resources and economy (Tabari, 2020; Pei et al., 2021). Accordingly, we include absolute indices such as the maximum 1 day precipitation (Rx1day) and total precipitation (PRCPTOT), a threshold-based index (number of heavy rain days, R10mm), percentile indices (e.g., total annual precipitation from very heavy rain days, R90p and R99p), and duration indices such as the consecutive wet (CWD) and dry days (CDD)."

2. L187: Out of curiosity, why choosing: question out of curiosity R99p + R90p, rather than R95p? Same question for Rx1day and Rx5days.

Responses: There was no special reason of not choosing R95p, basically we wanted to present a moderately extreme index (R90p) and a highly extreme index (R99p). We agree that Rx5days is also an important index. We presented the Rx1day because it may be more relevant to short duration impacts such as flash flood.

3. Regridding of datasets: it is a good idea to show the same figures on different resolution. Did did you also aggregate BARRA so that you could compare it to ERA5?

Responses: Actually, we have also aggregated BARRA to the ERA5 grid. Just that because AGCD is treated as observations here, we have mainly presented the re-gridded results to the AGCD grid in the Supplementary Information such that there is a common ground for comparison.

4. Regridding of datasets: as mentioned above, the method to regrid the 12 and 30 km grid to a 5 km grid is named but no details given in supplementary or in methods. To reproduce the analysis it would be useful to have some more details.

Responses: Thank for your reminding us on this. Bilinear interpolation was applied for re-gridding and we have added this information to the methodology section in the main text as well as the Supplementary Information. The revised statement in paragraph L192–198 is "The extreme indices calculated from reanalysis data have also been regridded to the 5-km resolution using bilinear interpolation, which are included in the supplementary information to demonstrate that our conclusions are insensitive to the choice of evaluation resolution."

*L199 - Evaluation matrices paragraph*

What is calculated and how is rather unclear, at least to me. 4 criteria of evaluation are mentioned line 200: "climatology, coefficient of variation (CV), temporal correlation, and trends".

- Does "climatology" refer to long-term mean? What is long-term (20 years, 30 years)?

Responses: This has been revised to "climatology (29 years in our case)" in the statement.

- I assume that R1xday, R10mm, R99p, R90p, CDD and CWD are calculated for every year within the studied period. The coefficient of variation, temporal correlation, and trend (which I assume is the slope of the variation of the index from year 1 to year x) are then calculated for every grid cell. Then, the value of CV, temporal correlation, and trend of each gridcell in ERA5 and BARRA are compared to the corresponding value in AGCD: this gives the bias. Finally, the average bias is computed for CV, temporal correlation, and trend in ERA5 and BARRA, which gives the domain-averaged bias. Could you confirm that this is correct?

Responses: The method you described was what we have performed.

I would suggest that this paragraph includes a table or a figure showing which method is applied to each of the 7 indices (6 climate extremes + PCRPTOT), as well as some more details on the methods. In particular, CV and temporal correlation are explained in one sentence, but there is no details for "climatology" and "trend".

Response: We think it would be clearer if we revise this paragraph and add more explanations about computation of the metrics, as in the following:

"We evaluate BARRA and ERA5 for their performance in capturing mean climatology (29 years in our case), coefficient of variation (CV), temporal correlation, and trends of the seven selected precipitation extreme indices. The CV is a valuable statistical tool representing the ratio of the (yearly) standard deviation to the mean, allowing for the comparison of variation between different data series, even when their means differ significantly. Temporal correlations, which are computed on yearly timescale, of climate extremes measure the similarities between simulations

and observations in terms of their inter-205 annual variabilities, with larger temporal correlations indicating better performance. For trend analysis, we applied simple linear trend line fitting to the yearly time series of climate indices. All the above metrics are computed at each grid point in the datasets' native grids as well as the AGCD grid after re-gridding. Differences between BARRA/ERA5 and AGCD then form the bias maps. After averaging over all grid points, the domain-averages will then be discussed in the following."

The time period must be specified for all climate indices.

Responses: As in our earlier response, the analysis time period (1990–2018) has been added to this section.

*Results*

L229: the bias in % would then be the "domain averaged relative bias"?

Responses: Yes, that is the domain averaged relative bias. The statement has been revised to "Domain averaged absolute bias in annual precipitation is about 0.17mm/day (~12.7% relative bias with respect to domain average) for BARRA and 0.15 mm/day (~10.5% relative bias) for ERA5 (Table 2)."

L233/Figure 2: the graph does not allow to see the 0.85 limit. Does this value come from another graph/a mean value calculated by region?

Responses: We have added a contour of value 0.85 to Figure 2, as shown in the following.

[Figure]

Figure 2 Temporal Correlation of annual precipitation between BARRA/ERA5 and AGCD. The contour (black) at value 0.85 is shown for reference.

L243: Tasmania seems to have low CV too?

Responses: Thanks for correcting this. The statement has been revised to "In the observation, CV is generally smaller for coastal regions including Tasmania except for northwest West Australia than inland Australia, where annual rainfall is much smaller than coastal regions."

L257: "reasonably well" lack a criterion. Maybe "roughly" as in line 267, or insist on the "similar pattern"? I would assume that the criterion is only visual.

Responses: We admit that "reasonably well" lacks clear criterion. The statement has been revised to "Both BARRA and ERA5 roughly reproduce the major trend pattern, however, clear biases can be observed over Northern Australia …".

L311-320: What about consecutive dry days?

Responses: We did miss out CDD in the paragraph. The statement has been revised to "Temporal correlation for CDD, CWD and R99p are not as good as R10mm and R90p. CDD has more regions with stronger correlations (0.5-0.6) or above than CWD and Rx1day, for the latter correlation is about ~0.5 or less over most of the domain."

*Conclusion*

L358-359: would be worth concluding with which climate extremes align and which differ

Responses: The statement has been revised to "Both BARRA and ERA5 align in CV patterns and biases for certain extremes (CWD, R10mm, R90p) but differ notably in others (mean climatology, trend, CDD, R99p, Rx1day)."

L376: "certain precipitation extremes" -> which ones? Can it be mostly duration-related? Or, if there is no link, just use the abbreviations or full names.

Responses: Thanks for pointing out this was not clear. Actually, these are not mostly the duration-related. Among the climate extremes CWD and R90p have less biases. The statement has been revised to "The results indicate that both BARRA and ERA5 demonstrate reasonable skill in simulating mean precipitation and certain precipitation extremes (e.g., CWD and R90p)."

L466: "a few indices": again, which ones?

Responses: At L446? The statement has been revised to "When examining temporal correlations for extreme precipitation indices compared to mean annual precipitation, both BARRA and ERA5 show weaker correlations, except for a few indices (CDD, R10mm, Rx1day) where ERA5 slightly outperforms BARRA."

**Technical corrections**

L183-191: Only 1 index is chosen in the "threshold-based" category, as well as in the "percentile indices" category. Can be rephrased so that it is clearer that only one index is within that category (avoid "such as", "e.g." etc. if there's only 1 choice).

Responses: This paragraph has been revised, see our response to the next comment.

L189: The same indices are chosen for the 2$^{nd}$ reason as for the 1$^{st}$. This sentence could be rephrased to show this. Ex: "Rx1day, CDD and CWD also capture the extremes that have the most impact on society and infrastructure …". The way it is described now let the reader think that the 6 core indices were chosen as "key aspects of climate extremes" and 2 of them spontaneously fitted in "extremes that have an impact on society and infrastructure". Is that the case?

Responses: We are sorry about this confusion. Actually, all seven indices have impacts on society and infrastructure. Thus, we have revised the paragraph as (also in response to Methodology comment 1):

"Although ClimPACT generates 14 precipitation-related core indices, we select seven (Table 1) based on the following considerations: 1) To capture key aspects of climate extremes, and 2) to capture extremes which have impacts on society and infrastructure such as agriculture, water resources and economy (Tabari, 2020; Pei et al., 2021). Accordingly, we include absolute indices such as the maximum 1 day precipitation (Rx1day) and total precipitation (PRCPTOT), a threshold-based index (number of heavy rain days, R10mm), percentile indices (e.g., total annual precipitation from very heavy rain days, R90p and R99p), and duration indices such as the consecutive wet (CWD) and dry days (CDD)."

L193: The sentence let the reader think that the "six precipitation-related indices" are not the same as those presented above.

Responses: This statement has been revised to "…the seven aforementioned extreme indices were calculated on native grids and observation grids."

L238: "large than" should be replaced by "larger than"

Responses: This has been corrected.

L259: The authors indicate a relative bias while figure 4 seems to shows absolute bias. Maybe harmonize?

Responses: This statement has been revised to "…both BARRA and ERA5 underestimate biases more than 100% (i.e., trend of 0.08 mm/day per year with bias of similar magnitude)".

L278: Please remind the reader which are the "precipitation related extremes", at least which ones are the "duration-related extremes". This would avoids going back to Table 1.

Responses: We have revised the statement to "duration-related extremes (CDD and CWD)" and "precipitation-related extremes (PRCPTOT, R10mm, R90p, R99p, Rx1day)".

L284: Since the bias goes up to 40%, it would be interesting to have a break in the legend in Figure 5 corresponding to that number.

Responses: The reviewer refers to the panels on Rx1day in Figure 5? We have added a contour at value of 40 in the panels to mark its location, as shown in the following.

[Figure]

Figure 5 Biases in Rx1Day in BARRA (left) and ERA5 (right). Stippling indicate areas with biases that are statistically significant at 95% confidence level.

L292-293: harmonize appellations: "heavy rainfall days" in the text and "heavy precipitation days" in the figure

Responses: This has been corrected.

L307: ".  In contrast," instead of ", in contrast,"

Responses: This has been corrected.

L314: "R90p": when was that introduced???

Responses: Introduction of "R90p" has been added back to paragraph L183–L191.

L348: "rx1day" à "Rx1day"

Responses: This has been corrected.

L352: The authors use of the abbreviation "CDD" and the long name "heavy rainfall event" in the same sentence. It would be clearer to choose one or the other.

Responses: This has been corrected.

**Citation**: https://doi.org/10.5194/hess-2024-286-RC2